# Evaluation of pseudo-healthy reconstruction for anomaly detection in brain FDG PET

**Ravi Hassanaly**                                        RAVI.HASSANALY@ICM-INSTITUTE.ORG
**Camille Brianceau**                              CAMILLE.BRIANCEAU@ICM-INSTITUTE.ORG
**Maëlys Solal**                                          MAELYS.SOLAL@ICM-INSTITUTE.ORG
**Olivier Colliot**                                                OLIVIER.COLLIOT@CNRS.FR
**Ninon Burgos**                                                   NINON.BURGOS@CNRS.FR
*Sorbonne Université, Institut du Cerveau - Paris Brain Institute - ICM, CNRS, Inria, Inserm, AP-HP, Hôpital de la Pitié Salpêtrière, F-75013, Paris, France*

**Editors:** Under Review for MIDL 2024

## Abstract

We propose an evaluation procedure based on the simulation of realistic abnormal images to validate pseudo-healthy reconstruction methods when no ground truth is available. We apply this framework to the reconstruction of 3D brain FDG PET using a convolutional variational autoencoder. This work has recently been published at MELBA.

**Keywords:** Deep learning, Pseudo-healthy reconstruction, Unsupervised anomaly detection, Variational autoencoder, 3D PET, Alzheimer's disease

## 1. Introduction

Unsupervised anomaly detection (UAD) using deep generative models (Chen and Konukoglu, 2022; Zhang et al., 2023) is an active field of research in medical imaging, especially in neuroimaging, where it has been widely applied to the detection of tumors and white matter hyper-intensities on structural magnetic resonance images (MRI) (Baur et al., 2021). We here focus on brain $^{18}$F-fluorodeoxyglucose (FDG) positron emission tomography (PET), a relevant modality to study Alzheimer's disease (AD), as it allows imaging brain areas with hypometabolism (Herholz, 1995). However, when applying UAD in this context (Choi et al., 2019; Baydargil et al., 2021), we do not have access to ground truth masks of the anomalies to evaluate the models.

We introduce a framework for the evaluation of pseudo-healthy reconstruction approaches in the absence of ground truth. It consists in simulating anomalies on images of healthy subjects to generate pairs of pathology-free and pathological (e.g., mimicking dementia-like lesions) images. We complement the framework by defining a new healthiness metric that measures whether the reconstructed image is of healthy appearance. We finally evaluate a VAE (Kingma and Welling, 2014) trained on 3D brain PET using the framework.

## 2. Methods

In addition to evaluating the quality of the generated images using reconstruction metrics that are common in the image synthesis literature (Nečasová et al., 2022), there is a need to evaluate if the reconstructions are looking healthy and allow detecting anomalies.

In most of the cases, datasets include lesion masks used as ground truth. However, for neurodegenerative disease studies using FDG PET, these masks are not available.

We propose to simulate hypometabolism on healthy images to have pairs of healthy (considered as ground truth) and abnormal images. We designed a mask corresponding to regions associated with AD (parietal and temporal lobes) (Landau et al., 2012). We then reduced the intensity of the PET signal within the region defined by the mask by different factors to simulate various degrees of hypometabolism as illustrated in Figure 1.

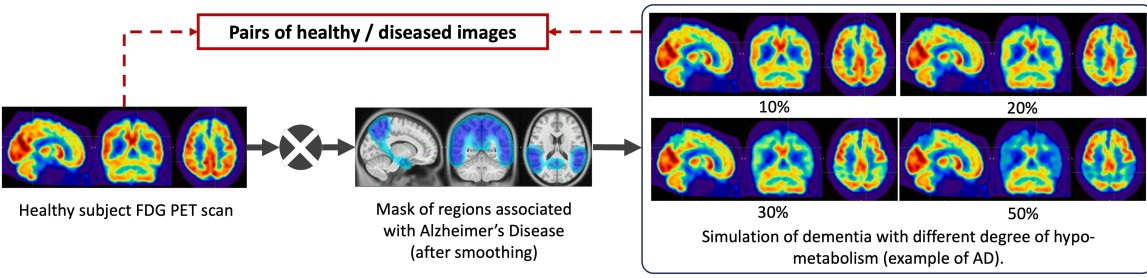

Figure 1: Hypometabolism simulation pipeline. The intensity of the image from a healthy subject is reduced by a chosen factor in a region associated with a dementia.

Furthermore, we define the "healthiness" metric $H = \frac{\mu_M}{\mu_{\bar{M}}}$ to measure if the model is able to reconstruct images that are looking healthy. $\mu_M$ is the average uptake in the region of the mask $M$ used to simulate the anomaly, and $\mu_{\bar{M}}$ is the average uptake of voxels in the brain excluding the mask $M$.

## 3. Experiments and results

FDG PET scans used in this study were obtained from the ADNI database (Jagust et al., 2010, 2015). After curating the dataset, we have 739 images from 378 cognitively normal (CN) subjects. We use the 60 baseline sessions from 60 CN subjects used for the test set to build new test sets by using our simulation method with different hypometabolism intensity degrees, resulting in a total of 8 simulated test sets.

We use a 3D convolutional VAE as VAEs have already shown their efficacy for UAD in medical imaging (Baur et al., 2021; Chen and Konukoglu, 2022).

We observe in Figure 2 that the input and output images of the CN subject are quite similar, both the shape of the brain and the uptake distribution look alike. The differences are due to the model imperfect reconstruction and correspond to the minimal error that it can achieve. When feeding the simulated hypometabolic image $\mathbf{x}'$ to the model, we observe that the reconstructed image $\widehat{\mathbf{x}'}$ looks healthier than the input image. The areas highlighted in blue in the residual map correspond to the regions where hypometabolism was simulated.

We can see in Figure 3 that the healthiness of simulated images $\mathbf{x}'$ is lower than for PET scans from CN subjects (the ground truth). However, the healthiness score of the reconstruction $\widehat{\mathbf{x}'}$ is always superior to the one of the simulated image $\mathbf{x}'$. We can see that it is even really close to the healthiness of the original image $\mathbf{x}$.

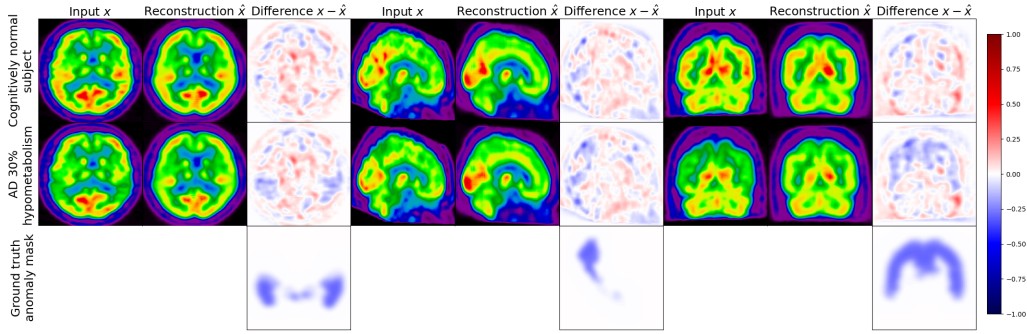

Figure 2: Results obtained from a real image of a CN subject (top row) and an image simulating AD hypometabolism based on the same CN subject (bottom row).

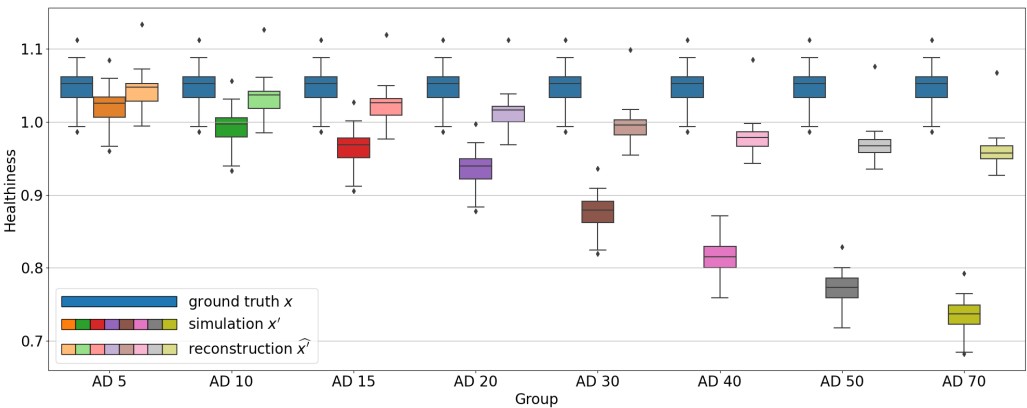

Figure 3: Distribution of the healthiness metric for the ground truth healthy images, their corresponding simulated images and their pseudo-healthy reconstructions when increasing the percentage of AD-like simulated hypometabolism.

## 4. Discussion and conclusion

To overcome the absence of ground truth for the evaluation of the model, we introduced a framework to simulate dementias related diseases from images of healthy subjects. This framework has been applied to a 3D VAE that is suited to detect anomalies due to dementia on brain FDG PET. Thanks to simulated test sets and the defined healthiness metric, we could evaluate the model ability to reconstruct healthy looking images. This work has recently been published at MELBA (Hassanaly et al., 2024).

## Acknowledgments

Funding was received from ANR (ANR-19-P3IA-0001, ANR-10-IAIHU-06). This work was granted access to to IDRIS HPC resources (AD011011648) made by GENCI.

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
