# OpenReview forum: "Evaluation of pseudo-healthy reconstruction for anomaly detection in brain FDG PET"
_MIDL.io/2024/Short_Papers — MIDL 2024 Short Papers_

### Official Review · Reviewer_Wejt · 2024-04-24

**Confidence:** 4
**Final Rating:** 5

**Review:**

In this paper, the authors present a framework to evaluate pseudo-healthy reconstruction approaches in the absence of ground truth. They simulate anomalies (hypometabolism) on healthy FDG PET images to have pairs of healthy and pathological images. They also introduce a metric to measure the healthiness of images when using the simulation framework. This procedure has been applied to a  VAE that is suited to detect anomalies due to dementia on brain FDG PET.
The presentation of the paper is clear and well-written. This work has recently been published at MELBA.

---

### Decision · Program_Chairs · 2024-04-26

Accept